# Increased adverse events following third dose of BNT162b2/Pfizer vaccine in those with previous COVID-19, but not with concurrent influenza vaccine

**Rachael K. Raw** [1]\*, **Jon Rees** [2], **David R. Chadwick** [3]

**1** School of Medicine and Health, Newcastle University, Newcastle upon Tyne, United Kingdom, **2** The School of Psychology, University of Sunderland, Sunderland, United Kingdom, **3** Centre for Clinical Infection, James Cook University Hospital, Middlesbrough, United Kingdom

\* Rachael.raw3@nhs.net

**Data Availability Statement:** Data is uploaded as supplementary information.

**Funding:** The CHOIS study was supported by the North East and North Cumbria Academic Health

## Abstract

Prior studies suggest that adverse events (AEs) following doses one and two of BNT162b2/Pfizer vaccine are worse in those with a prior history of COVID-19. To establish whether this outcome applies to a third/booster dose, we conducted a survey with 534 healthcare workers (HCW) in Northeast England, who reported AEs following all three doses of BNT162b2/Pfizer vaccine. We also explored AEs associated with concurrent seasonal influenza immunisation, in a subset of 492 HCWs. For all doses of BNT162b2/Pfizer vaccine there was a cluster of systemic AEs that were consistently worse in HCWs with a prior history of COVID-19. AEs were no worse in HCWs who received their third/booster dose within 7 days of the influenza jab, rather than further apart. Gender and the presence of ongoing COVID-19 symptoms (OCS) had no effect on AEs associated with COVID-19 or influenza vaccination, though younger HCWs experienced more AEs overall. Our findings have implications for vaccine hesitancy and immunisation protocols.

## 1. Introduction

Global vaccination efforts against SARS-CoV-2 began in late 2020, with BNT162b2/Pfizer [1], mRNA-1273/Moderna [2], and ChAdOx1/AstraZeneca [3] vaccines applied predominantly in the UK population. Healthcare Workers (HCWs) have been a priority target group. Phase three trials demonstrated general safety of these vaccines, but adverse events (AEs) have been widely reported, albeit mostly mild-to-moderate in nature and short in duration [4–10]. We have previously demonstrated an association between prior COVID-19 infection and increased AEs following administration of dose one of BNT162b2/Pfizer vaccine, but not after dose two [9, 10]. The presence of "ongoing COVID-19 symptoms" (OCS; symptoms persisting from 4–12 weeks) [11] was also associated with increased AEs after the second dose of BNT162b2/Pfizer vaccine [9, 10].

In late 2021, a 'booster' dose of COVID-19 vaccine was recommended, and as of May 2022, just under 40 million UK citizens had received this third dose [12]. To our knowledge, it is

Sciences Network (AHSN; fund awarded to co-author DRC). The funders had no role in study design, data collection and analysis, decision to publish, or preparation of the manuscript.

unknown whether the previously observed effects of factors such as COVID-19 history carry over to the booster dose. Furthermore, due to the timing of booster vaccinations, many eligible individuals, particularly HCWs, were invited to receive their COVID-19 booster at the same time as their seasonal influenza jab, which raises the question of whether concomitant influenza/COVID-19 vaccination may enhance AEs. Large trials have shown that co-administration of influenza vaccine with the second dose of a COVID-19 vaccination (including NVX-CoV2373 [13], ChAdOx1 [14] or BNT162b2 [14]) did not yield any worse AEs than when the COVID-19 vaccine was delivered on its own. No study has yet looked at concomitant administration in the context of the COVID-19 third/booster dose.

The primary aim of our present study was to explore the impact of risk factors for vaccine-associated AEs following the third/booster dose of BNT162b2/Pfizer vaccine in HCWs. This included prior COVID-19 history, the presence of OCS, and HCWs' age/gender. We also examined whether prior COVID-19 history would lead to worse AEs following the influenza jab. As a secondary aim, in a small subgroup, we also compared AEs between those who received their BNT162b2/Pfizer booster around the time of their Influenza jab, and HCWs who had their vaccinations > 7 days apart.

## 2. Method

### 2.1. Ethics statement

All participants gave their written informed consent, and Cambridge East Research Ethics Committee approved this study (Ethics Ref: 20/EE/0161), which was performed in accordance with the ethical standards laid down in the 1964 Declaration of Helsinki.

### 2.2. Participants

HCWs recruited from three Northeast England hospitals, formed an opportunistic sample for this retrospective observational and descriptive study. HCWs were invited via email, to complete an anonymous online survey, which captured their experience of AEs following first, second and third/booster doses of the BNT162b2/Pfizer vaccine using structured and multiple-choice questions The email was sent out to all HCWs, both clinical and non-clinical, using the weekly Hospital Communications service. Those who had received all three doses of BNT162b2/Pfizer vaccine were eligible to partake.

### 2.3. Materials and procedure

AEs were documented using a modified version of the FDA Toxicity Grading Scale. [15] as previously described [9, 10]. HCWs indicated whether they had received influenza vaccine within 7 days of the COVID-19 booster dose. If HCWs did not receive both vaccinations within this time interval, they were also asked to describe AEs following the influenza vaccine. Prior COVID-19 history was classed as positive for HCWs with self-reported positive PCR and/or COVID-19 test(s). The number of moderate-to-severe AEs persisting >24hr were calculated for doses one, two and three of BNT162b2/Pfizer, and for influenza where it was not administered concomitantly. For BNT162b2/Pfizer doses one and two, AEs were only counted as moderate/severe if they were rated as "similar or worse than my current symptoms". Effects of age, gender, presence of OCS, and whether influenza vaccine was administered concomitantly were also considered.

## 2.4. Statistical analysis

Statistical analysis was conducted using JASPv0.16.3.0 [16]. Frequencies of categorical variables and the occurrence of any moderate-to-severe AEs were examined using the chi-square test. Scores were compared using 2-way ANOVA and ANCOVA where age and gender were relevant. Multivariable logistic regressions were applied to identify the relationship between COVID-19 status and moderate-to-severe AEs, and the Bonferroni correction applied to the resulting significance/confidence intervals. Correlation/tests of difference explored associations between AEs and age/gender.

## 3. Results

Five-hundred and thirty-four HCWs responded to the questionnaires (15% male, mean age 49.6yr [SD-10.6]. Looking at prior COVID-19 history, 147/454 (32.3%) self-reported previous COVID-19. 492 (92.1%) had received influenza vaccine: 56 HCWs (11.4%) had it within 7 days of their third BNT162b2/Pfizer dose, 292 HCWs (59.4%) had their doses >7 days apart. 144 HCWs (29.3%) did not indicate time interval. Raw data is available to view as supplementary information (see S1 Data).

Initial analyses revealed a significant negative relationship between age and the number of moderate-to-severe AEs following third BNT162b2/Pfizer and influenza vaccines (BNT162b2/Pfizer Kendall's-tau = -0.16; influenza Kendall's-tau = -0.14;both p<0.001). However, for BNT162b2/Pfizer there was no significant relationship between age and number of AEs reported following doses one and two. There was a significant strong positive relationship between number of AEs reported for doses one and two (Kendall's-tau = 0.46;p<0.001), a negative relationship between AEs following doses one and three (Kendall's-tau = -0.13;p<0.001), but no relationship between number of AEs for doses two or three, or between each of doses two and three, and the number of influenza AEs reported. There were no significant differences in number of AEs reported by male versus female HCWs, as shown in Table 1.

For occurrence of one or more moderate/severe AEs, there was a significant association with previous COVID-19 and BNT162b2/Pfizer dose one (O.R.-1.64[1.13–2.38];p = 0.011), dose two (O.R.-1.63[1.12–2.35];p = 0.011), dose three (O.R.-2.44[1.67–3.55];p<0.001), and between previous history of COVID-19 and occurrence of AEs following the influenza vaccine (O.R.-1.88[1.25–2.82];p = 0.002). To investigate the effect of prior COVID-19 on number of AEs following all three BNT162b2/Pfizer vaccines, a two-by-three (AEs x vaccine dose) ANOVA was performed. Age and gender were not used as covariates here given the non-significance of relationships identified with doses one and two, as described above. There was an overall main effect of prior COVID-19 (F(1.8, 957.7) = 48.63;p<0.001), whereby HCWs with prior COVID-19 reported 2.59 [2.39–2.79] versus 1.91[1.71–2.12] AEs (Fig 1A). There was a significant difference in number of AEs reported following dose one (F(1,527) = 19.93; p<0.001), with post-hoc testing (Bonferroni-p<0.05) demonstrating that the number of AEs

**Table 1. Number of AEs by participant gender and vaccine dose: Mean (S.D) number of AEs reported by male versus female HCWs for all three BNT162b2/Pfizer doses and the seasonal Influenza vaccine.**

|  | Male (n = 80) | Female (n = 454) | p | d [95% C.I.] |
|---|---|---|---|---|
| Age | 48.7 (11.5) | 49.6 (10.7) | - | - |
| BNT162b2/Pfizer Dose 1 | 2.84 (2.85) | 2.86 (2.75) | 0.95 | 0.008 [-0.23–0.25] |
| BNT162b2/Pfizer Dose 2 | 2.44 (2.90) | 2.55 (2.80) | 0.74 | 0.04 [-0.2–0.28] |
| BNT162b2/Pfizer Dose 3 | 0.99 (1.97) | 1.43 (2.01) | 0.07 | 0.22 [-0.02–0.46] |
| Influenza | 0.78 (1.79) | 0.76 (1.48) | 0.90 | - 0.02 [-0.26–0.23] |

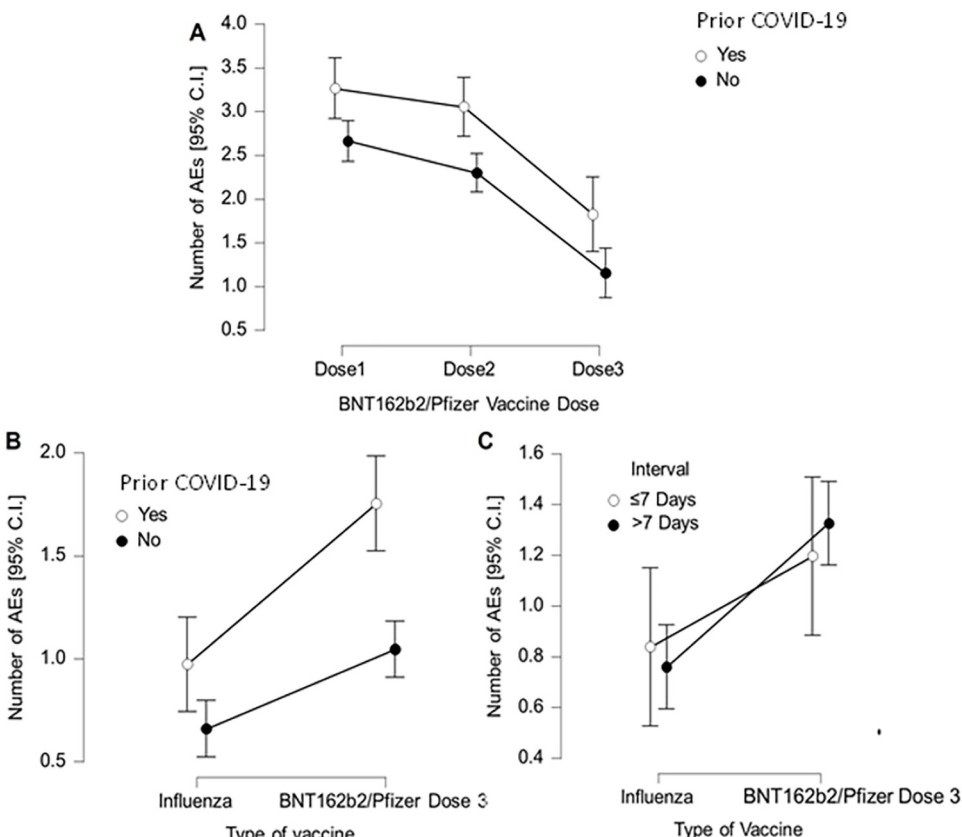

**Fig 1. Frequency of AEs by vaccine type, prior COVID-19 status and interval between doses. A.** Number of AEs according to previous COVID-19 history, for all three BNT162b2/Pfizer doses. **B.** Number of AEs according to previous COVID-19 history for BNT162b2/Pfizer third/booster dose and the seasonal influenza vaccine. **C.** Number of AEs according associated with the influenza vaccine versus BNT162b2/Pfizer third/booster dose for those HCWs who had these vaccines ≤ or > 7 days apart.

reported for dose three (1.36 [1.14–1.59]) was significantly lower than reported for dose one (2.84[2.61–3.07]), and dose two (2.55[2.32–2.78]. There was no interaction between vaccine dose and COVID-19 history (Fig 1A).

To investigate the difference in number of AEs between BNT162b2/Pfizer dose three and the influenza vaccine, together with the role of interval between each vaccine, a two-by-two-way ANCOVA (vaccine x dosage interval), with age and gender as covariates (given the significant relationships here with dose three), was performed. Overall, there were significantly more AEs following BNT162b2/Pfizer dose three (1.28[1.07–1.49]), compared to the influenza vaccine (0.81[0.60–1.02]; Fig 1B); AEs $F_{(1,331)} = 3.86$;d = 0.28[0.19–0.37];p<0.001), but there was no significant effect of interval between vaccines ($F_{(1,331)} = 0.02$;p = 0.88), and no interaction between these variables ($F_{(1,311)} = 0.61$;p = 0.44; Fig 1C).

Given the association between occurrence of any influenza AEs and prior COVID-19, we investigated this further. A two-by-two-way ANCOVA (vaccine-type x prior COVID-19) was performed, with age and gender as covariates. The overall significant difference in number of AEs between prior COVID-19 and influenza was maintained ($F_{(1,470)} = 1.06$;p<0.001), but there was also a significant overall effect of prior COVID-19 with those previously symptomatic reporting more AEs across both types of vaccine ($F_{(1,470)} = 11.73$,p<0.001), but crucially a significant interaction between vaccine type and COVID-19 history $F_{(1,470)} = 4.16$;

**Table 2. Moderate and severe AEs by COVID-19 status: Percentage of cases reporting moderate or severe AEs following BNT162b2/Pfizer third/booster dose (95% CI) in those with and without a history of COVID-19 (the former including OSC).**

|  | Dose 3 | |
|---|---|---|
|  | **Odds Ratio (95% C.I.)** | ***p*** |
| **Fever** | 1.44 (0.55 3.81) | >.99 |
| **Fatigue** | 2.37 (1.32 4.27) | < .001 |
| **Myalgia** | 2.25 (1.18 4.28) | 0.001 |
| **Arthralgia** | 2.10 (0.98 4.46) | 0.06 |
| **Lymphadenopathy** | 0.79 (0.79 1.40) | >.99 |
| **Local Pain** | 1.43 (0.77 2.65) | >.99 |
| **Local Redness** | 2.56 (0.77 8.53) | 0.33 |
| **Local Swelling** | 3.15 (0.82 12.11) | 0.18 |
| **Nausia & Vomiting** | 0.51 (0.12 2.12) | >.99 |
| **Diarrhoea** | 1.49 (0.23 9.47) | >.99 |
| **Headache** | 2.03 (1.10 3.77) | 0.011 |

$p = 0.04$). A simple effects analysis revealed this interaction was due to significantly larger numbers of AEs to BNT162b2/Pfizer vaccine in those with prior COVID-19 ($t(527) = 3.61$, $p<0.001; d = 0.34[0.15–0.52]$), whilst the difference in number of AEs following influenza vaccine was not significant ($t(488) = 1.78; p = 0.08; d = 0.17[-0.02–0.36]$).

A similar pattern of results was found for prolonged AEs, which were scored as total number of vaccine-associated AEs rated as moderate/severe, lasting >24 hours. In this instance, an ANCOVA found significantly higher numbers of prolonged AEs associated with BNT162b2/Pfizer vaccine compared to influenza ($1.09[0.96–1.23]$ v $0.32[0.19–0.45]$; AEs $F(1,511) = 15.24$; $p<0.001$), a higher number of prolonged AEs in those with prior COVID-19, than without ($0.84[0.69–1.00]$ v $0.57[0.42–0.72]$; AEs ($F(1,511) = 4.09; p = 0.044$), and a significant interaction between these variables ($F(1,511)–7.13; p = 0.008$). A simple effects analysis showed that the number of prolonged moderate-to-severe AEs was again only significantly higher following BNT162b2/Pfizer vaccine ($t(528) = 3.03; d = 0.28[0.10–0.47]; p = 0.003$), with no significant difference in prolonged AEs for the influenza vaccine ($t(532) = 0.44; d = 0.04[-0.14–0.22]$; $p = 0.66$).

Incidence of each type of AE following BNT162b2/Pfizer dose three for those with and without prior COVID-19, is shown in Table 2. As in our previous work, the cluster of headache, fatigue, myalgia and arthralgia were most common in HCWs with prior COVID-19. Overall, the incidence of AEs to dose three were remarkably like those following the first two doses. Looking at the types of AEs most frequently reported, headache, fatigue, and myalgia were associated with prior COVID-19 for the third/booster dose of BNT162b2/Pfizer, after correcting for multiple comparisons, age and gender (Fig 2; Table 2). The effect for arthralgia was very nearly significant ($p = 0.06$). A similar analysis for time interval between influenza and BNT162b2/Pfizer dose three, found no significant relationship for occurrence of any of the vaccine-associated AEs (all $p>0.9$).

Comparing the subset of HCWs with OCS (n = 32), with those without OCS, there was a similar reduction in number of AEs to BNT162b2/Pfizer dose three, compared to doses one and two ($F(1.82,957.8) = 11.89; p<0.001$; dose one $2.91[2.41–3.4]$; dose two $2.48[1.97–2.98]$; dose three $1.44[1.08–1.80]$; Bonferroni-corrected post-hoc for dose three being significantly different [$p<0.05$] to doses one and two. There was no significant overall difference in number of AEs between groups ($F(1,527) = 0.03; p = 0.86$ (OCS mean $2.30[1.73–2.87]$; control mean

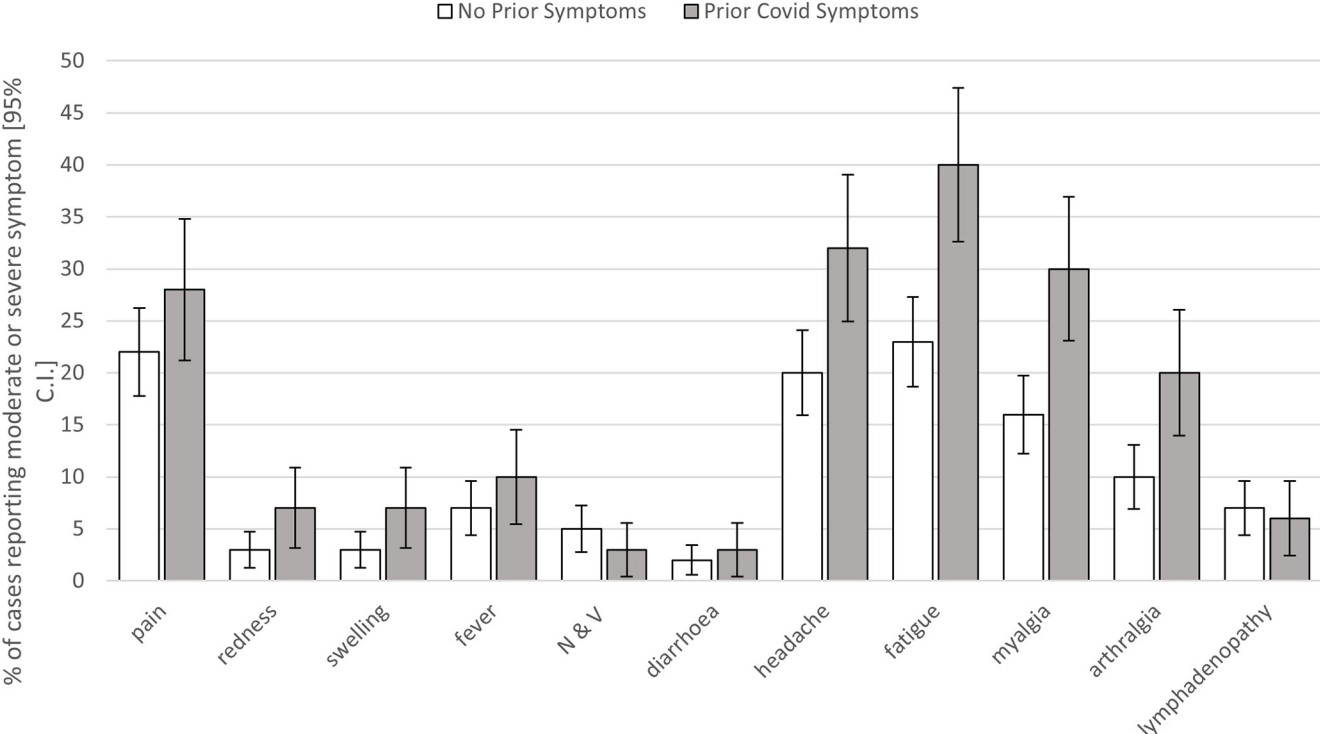

**Fig 2. Moderate and severe AEs by COVID-19 status: Percentage of cases reporting moderate or severe AEs (95% CI) following BNT162b2/Pfizer third/booster dose in those with and without a history of COVID-19.** N & V: nausea and vomiting.

2.25[2.10–2.39]), and no significant interaction (F(1.82, 957.8) = 0.15;p = 0.84). This suggests that OCS had no effect on number of AEs reported at each BNT162b2/Pfizer dose interval. Furthermore, there was no significant difference in number of AEs experienced following the influenza jab, between those with versus without OCS (without = 0.70;SD-1.5; with = 0.56,SD-1.4; AEs d = 0.09[-0.30–0.56];p = 0.54).

## 4. Discussion

Our previous work highlighted risk factors for increased BNT162b2/Pfizer vaccine-associated AEs, specifically females, younger age, prior COVID-19, and OCS were associated with increased AEs reported following doses one and/or two [9, 10]. The main findings of our present study were firstly, that frequency of AEs to BNT162b2/Pfizer vaccination progressively reduced across all doses. Nevertheless, moderate-to-severe AEs were markedly more common in our cohort compared to the COV-BOOST trial that used the same AE grading system following third/booster doses of BNT162b2/Pfizer (AEs<5%;n = 96) [17]. Our finding of progressive reductions in AEs across successive doses contrasts with both our prior work (showing AEs increased between doses one and two [10]) and a recent Israeli stud showing the opposite effect [18]. This might have occurred because either the present work involved a new sample of HCWs (hence individual differences) or HCWs becoming more accustomed to vaccine-associated AEs over time. Similarly, another surprising result was that gender did not appear to influence AEs, whereas our past studies have consistently shown females to report more AEs following doses one and two of BNT162b2/Pfizer vaccination [9, 10]. Younger participants in the present study reported more AEs following the third/booster BNT162b2/Pfizer

dose, and the influenza vaccine, the former finding being consistent with our and other's prior work [9, 10].

Our second key finding was that the effect of prior COVID-19 on vaccine-associated AEs, was again carried over to the third/booster BNT162b2/Pfizer dose in terms of increased AEs. Nevertheless, the presence of self-reported OCS did not seem to impact AEs to BNT162b2/Pfizer vaccination at any of the dosage timepoints, although our sample of HCWs with OCS was notably small.

Regarding the effect of influenza vaccination on AEs experienced around the time of the third/booster BNT162b2/Pfizer dose, our analysis was somewhat limited by the small number of HCWs that received their influenza vaccine within 7 days of t BNT162b2/Pfizer vaccination (n = 56). Analyses did however show that HCWs in this subgroup reported similar numbers/ levels of AEs compared to individuals who received vaccines more than 7 days apart. This supports previous safety data, which found no worse AEs reported following concomitant administration of influenza with doses one or two of COVID-19 vaccines [13, 14].

Limitations of the present work include the large sample imbalance for participant gender (85% female), and the retrospective recall (up to 12 months) of AEs in the present study. Given that vaccine safety plays a central role in people's decision to accept a vaccination, our present work has implications for vaccine hesitancy [19, 20]. We found that BNT162b2/Pfizer-related AEs diminished with each successive dose and were no worse than the AEs reported by HCWs following influenza vaccine, which has been widely accepted for many years. Our present work also provides preliminary data to support concomitant administration of the third/booster BNT162b2/Pfizer vaccination alongside the influenza vaccine. These findings of relatively mild AEs, even when given alongside influenza vaccine, should be used to bolster public confidence in immunisation programmes.

## 5. Conclusion

Prior history of COVID-19 was associated with increased AEs to BNT162b2/Pfizer vaccination at all three dosage points. The presence of OCS and female gender had no impact on AEs to BNT162b2/Pfizer or influenza vaccination, though younger HCWs experienced more AEs. Lastly, AEs were no worse in HCWs who received concomitant booster BNT162b2/Pfizer and influenza vaccines.

## Supporting information

**S1 Data. Raw data for all participants who received 3 doses of BNT162b2/Pfizer vaccine.** (CSV)

## Acknowledgments

We would like to thank the CHOIS research team, John Rouse and the North East and North Cumbria NIHR for assistance with the survey.

## Author Contributions

**Conceptualization:** Rachael K. Raw, Jon Rees, David R. Chadwick.

**Data curation:** Rachael K. Raw, Jon Rees.

**Formal analysis:** Jon Rees.

**Funding acquisition:** David R. Chadwick.

**Investigation:** Rachael K. Raw, David R. Chadwick.

**Methodology:** Rachael K. Raw, David R. Chadwick.

**Project administration:** Rachael K. Raw.

**Resources:** David R. Chadwick.

**Supervision:** David R. Chadwick.

**Validation:** Jon Rees.

**Writing – original draft:** Rachael K. Raw.

**Writing – review & editing:** Rachael K. Raw, Jon Rees, David R. Chadwick.

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
