## [Decision Letter · Decision Letter 0]

6 Dec 2022

PGPH-D-22-01340

Increased adverse events following third dose of BNT162b2/Pfizer vaccine in those with previous COVID-19, but not with concurrent influenza vaccine.

Dear Dr. Raw,

Thank you for submitting your manuscript to PLOS Global Public Health. After careful consideration, we feel that it has merit but does not fully meet PLOS Global Public Health’s publication criteria as it currently stands. Therefore, we invite you to submit a revised version of the manuscript that addresses the points raised during the review process.

We look forward to receiving your revised manuscript.

Kind regards,

Muhammad Imran Nisar, MBBS, MSc

Academic Editor

Journal Requirements:

2. Please send a completed 'Competing Interests' statement, including any COIs declared by your co-authors. If you have no competing interests to declare, please state "The authors have declared that no competing interests exist". Otherwise please declare all competing interests beginning with the statement "I have read the journal's policy and the authors of this manuscript have the following competing interests:"

3. Please amend your detailed Financial Disclosure statement. This is published with the article. It must therefore be completed in full sentences and contain the exact wording you wish to be published.

4. Please provide separate figure files in .tif or .eps format only and remove any figures embedded in your manuscript file. Please also ensure that all files are under our size limit of 10MB.

5. We have noticed that you have uploaded Supporting Information files, but you have not included a list of legends. Please add a full list of legends for your Supporting Information files after the references list. 

6. In the online submission form, you indicated that "All raw data will be made available upon request to the corresponding author". All PLOS journals now require all data underlying the findings described in their manuscript to be freely available to other researchers, either 1. In a public repository, 2. Within the manuscript itself, or 3. Uploaded as supplementary information.

Additional Editor Comments (if provided):

Reviewers' comments:

Reviewer's Responses to Questions

**Comments to the Author**

1. Does this manuscript meet PLOS Global Public Health’s publication criteria? Is the manuscript technically sound, and do the data support the conclusions? The manuscript must describe methodologically and ethically rigorous research with conclusions that are appropriately drawn based on the data presented.

Reviewer #1: Yes

Reviewer #2: Yes

Reviewer #3: Yes

2. Has the statistical analysis been performed appropriately and rigorously?

Reviewer #1: Yes

Reviewer #2: Yes

Reviewer #3: Yes

3. Have the authors made all data underlying the findings in their manuscript fully available (please refer to the Data Availability Statement at the start of the manuscript PDF file)?

Reviewer #1: Yes

Reviewer #2: Yes

Reviewer #3: No

4. Is the manuscript presented in an intelligible fashion and written in standard English?

Reviewer #1: Yes

Reviewer #2: Yes

Reviewer #3: Yes

5. Review Comments to the Author

Reviewer #1: Reviewer Comments:

The authors have a grandiose study, very important to health, as it tries to shade light on the implication for covid-19 vaccine uptake and immunization protocols. The study is very promising, but some minor points must be reviewed before possible publication.

Abstract:

The authors in the present study had a sample size of 534 HCW who were enrolled in the study:

The authors should be specific how many individuals were analysed separately for AEs following all three doses of BNT162b2/Pfizer vaccine and those were involved in the concurrent seasonal influenza immunisation.

Methods:

The authors needs to provide ethical approval number for this study

The authors should clearly demonstrate how they arrived at sample size of 534 (show sample size calculation)

Results

In line 156-157 the authors states that “A similar pattern of results was found for prolonged AEs, which were scored as total number of vaccine-associated AEs rated as moderate/severe, lasting >24 hours”. In the subsequent studies they could have performed correlation analysis between the adverse events within less than 24hrs and after prolonged period in order to paint a clear picture as to whether the development and increase in AEs is observed after long period of vaccination.

Conclusion

Since the large sample size that was analyzed constituted 85% female, can we conclusively state that AEs witnessed during covid-19 vaccination programs may be gender specific?

Reviewer #2: Reviewer’s comment

Introduction

(1)Line 41: Authors should indicate the exact AEs that were frequently reported

(2)Lines 43/46: It would be appropriate if authors differentiate or state the specific AEs reported following a single dose of BNT162b2/Pfizer vaccine and the events reported during ongoing COVID-19 symptoms vaccination with BNT162b2/Pfizer 47 vaccine after second dose.

Methods

(1)Lines 78-80: Provide the ethics approval reference number

(2)The methodology employed in this students has some gaps that need to be addressed or filled.

a.Authors employed the use of email to invite responses. However, it was not clear how the respondents were selected, no clear inclusion and exclusion criteria were stated.

b.Sample size was not predetermined.

c.The use of emails did not allow authors to verify the AEs being described. How were you able to verify these events?

d.Was the questions structured or unstructured?

e.Were there avenue for follow up, if you required further information

f.The use of emails did not allow anonymity. In this case, how did you ensured anonymity?

Results

(1)Lines 99/101: Avoid beginning a sentence with a number. Please revise

(2)Line 101: 144 HCWs (29.3%) did not indicate time interval. This seems a flaw in the design of the questionnaire. Authors should have envisaged this, even at the design stage.

Discussion

(1)Line 204: stud---study

(2)Our findings have implications for vaccine hesitancy and immunisation protocols. This was the concluding part of the abstract. The discussion section only has this on vaccine hesitancy ‘Given that vaccine safety plays a central role in people’s decision to accept a vaccination, our present work has implications for vaccine hesitancy’ It makes it very difficult to understand how this study could have implications on vaccine hesitancy and immunization protocols, in general. Please discuss the study to take care of this.

Reviewer #3: The authors are advised to include a statement about the type of study design that was employed.

Page4 line 88: The authors should provide citation for the software JASPv0.16.3.0.

Authors should avoid starting a sentence with abbreviations.

There are also a couple of spelling errors in the discussion aspect of the manuscript, the authors are advised to revise for punctuation and spelling.

6. PLOS authors have the option to publish the peer review history of their article (what does this mean?). If published, this will include your full peer review and any attached files.

**Do you want your identity to be public for this peer review?** For information about this choice, including consent withdrawal, please see our Privacy Policy.

Reviewer #1: **Yes: **JAMES NYABUGA NYARIKI (PhD)

Reviewer #2: No

Reviewer #3: No

---

## [Decision Letter · Decision Letter 1]

25 Jan 2023

Increased adverse events following third dose of BNT162b2/Pfizer vaccine in those with previous COVID-19, but not with concurrent influenza vaccine.

PGPH-D-22-01340R1

Dear Dr Raw,

We are pleased to inform you that your manuscript 'Increased adverse events following third dose of BNT162b2/Pfizer vaccine in those with previous COVID-19, but not with concurrent influenza vaccine.' has been provisionally accepted for publication in PLOS Global Public Health.

Best regards,

Abram L. Wagner, PhD, MPH

Academic Editor

Reviewer Comments (if any, and for reference):

Reviewer's Responses to Questions

**Comments to the Author**

1. If the authors have adequately addressed your comments raised in a previous round of review and you feel that this manuscript is now acceptable for publication, you may indicate that here to bypass the “Comments to the Author” section, enter your conflict of interest statement in the “Confidential to Editor” section, and submit your "Accept" recommendation.

Reviewer #1: All comments have been addressed

Reviewer #2: All comments have been addressed

Reviewer #3: All comments have been addressed

2. Does this manuscript meet PLOS Global Public Health’s publication criteria? Is the manuscript technically sound, and do the data support the conclusions? The manuscript must describe methodologically and ethically rigorous research with conclusions that are appropriately drawn based on the data presented.

Reviewer #1: Yes

Reviewer #2: Yes

Reviewer #3: Yes

3. Has the statistical analysis been performed appropriately and rigorously?

Reviewer #1: Yes

Reviewer #2: Yes

Reviewer #3: Yes

4. Have the authors made all data underlying the findings in their manuscript fully available (please refer to the Data Availability Statement at the start of the manuscript PDF file)?

Reviewer #1: Yes

Reviewer #2: Yes

Reviewer #3: Yes

5. Is the manuscript presented in an intelligible fashion and written in standard English?

Reviewer #1: Yes

Reviewer #2: Yes

Reviewer #3: Yes

6. Review Comments to the Author

Reviewer #1: All the issues I raised in the manuscript has been addressed to my satisfaction

Reviewer #2: All comments have been adequately addressed

Reviewer #3: The authors have satisfied all my concerns, the manuscript can now be accepted for publication.

7. PLOS authors have the option to publish the peer review history of their article (what does this mean?). If published, this will include your full peer review and any attached files.

**Do you want your identity to be public for this peer review?** For information about this choice, including consent withdrawal, please see our Privacy Policy.

Reviewer #1: **Yes: **Dr. JAMES NYABUGA NYARIKI

Reviewer #2: **Yes: **Enoch Aninagyei

Reviewer #3: No
